# Long-term physical and mental health outcomes of Ebola Virus Disease survivors in Kenema District, Sierra Leone: A cross-sectional survey

**Jenna N. Bates**[1]*, **Abdulai Kamara**[2], **Mohamed Sheku Bereteh**[2], **Denise Barrera**[3], **Lina Moses**[3], **Allieu Sheriff**[4], **Fudia Sesay**[4], **Mohamed S. Yillah**[4], **Donald S. Grant**[5,6], **Joseph Lamin**[2], **Philip Anglewicz**[7]*

1 School of Global Health, University of Copenhagen, Copenhagen, Denmark, 2 School of Community Health Sciences, Njala University, Bo, Sierra Leone, 3 Tulane University School of Public Health and Tropical Medicine, New Orleans, Louisiana, United States of America, 4 Sierra Leone Association of Ebola Survivors, Kenema, Sierra Leone, 5 Kenema Government Hospital, Kenema, Sierra Leone, 6 College of Medicine and Allied Health Sciences, University of Sierra Leone, Freetown, Sierra Leone, 7 Johns Hopkins Bloomberg School of Public Health, Baltimore, Maryland, United States of America

* jennab318@gmail.com (JNB); panglew@jhu.edu (PA)

**Data Availability Statement:** All relevant data are within the paper and Supporting Information files.

## Abstract

The 2013–2016 Ebola Virus Disease (EVD) epidemic in West Africa was the deadliest in history, with over 28,000 cases. Numerous physical and mental health symptoms have been reported in EVD survivors, although there is limited prior research on how the health of survivors compares to the general population. We conducted a survey of EVD survivors in Kenema District, Sierra Leone and a population-based sample of community members who lived in EVD-affected areas but were not diagnosed with EVD, and compared resulting data about self-reported symptoms, duration, and severity between EVD survivors and community members through multivariate regression models. This study found that more than six years after the epidemic, survivors were significantly more likely to experience both physical and mental health symptoms than community members, with respective adjusted incidence rate ratios (IRRs) of 2.65 (95% CI, 2.28–3.09), $p < 0.001$, and 11.95 (95% CI, 6.58–21.71), $p < 0.001$. The most highly reported physical health symptoms experienced by EVD survivors were joint pain (75.5%), headaches (67.3%), and vision problems (44.5%), and the most prevalent psychological symptoms were spells of terror and panic (25.5%) and difficulty falling asleep or staying asleep (20.0%). EVD survivors were significantly more likely than community members to report the symptoms as lasting for a longer period, a median of 6.0 (3.0–7.0) years, and with higher severity. The results indicated that six years after the epidemic, EVD survivors in Kenema District, Sierra Leone are experiencing worse physical and mental health than their peers. These findings of the long-term, debilitating health issues following EVD infection should be considered when designing and implementing future epidemic responses.

**Funding:** This work was funded by the NIH under grant number 1R21HD098504: The Impact of Ebola Infection on Demographic and Social Outcomes in Sierra Leone, received by PA and LM. The funders had no role in study design, data collection and analysis, decision to publish, or preparation of the manuscript.

**Competing interests:** The authors have declared that no competing interests exist.

## Introduction

Despite the prominence of Ebola Virus Disease (EVD), research on the long-term impacts on physical and mental health has important limitations. EVD is a severe illness that has impacted tens of thousands of individuals since its discovery in the Democratic Republic of Congo (DRC) in 1976. The 2013–2016 EVD epidemic in West Africa was the deadliest in history, with 28,652 suspected, probable, or confirmed cases [1]. It was largely centered in Liberia, Guinea, and Sierra Leone, but had a far reach with limited, quickly controlled cases in Nigeria, Mali, Senegal, the United States, the United Kingdom, and Spain. Sierra Leone was the most heavily impacted by the epidemic, with 14,124 cases (8,706 confirmed, 287 probable, and 5,131 suspected), 3,956 confirmed deaths, and more suspected [1, 2]. With a case fatality rate of 41% and a tendency to spread in intimate interpersonal activities such as caregiving and burial processes, EVD was highly destructive to communities and households throughout the affected regions [2, 3].

"Post-Ebola Syndrome" is used to describe the physical sequelae experienced by many survivors, with joint and muscle pain, headaches, and ocular symptoms being among the most commonly reported sequelae [4–9]. There are also several studies concerning the mental health of survivors, finding that they are at higher risk of developing depression, anxiety, and post-traumatic stress disorder (PTSD) [10–16]. This is consistent with findings from other epidemics and disaster scenarios, in which experiences during and after the event such as hospitalization, isolation, loss, having a life-threatening illness, and subsequent stigmatization, can all contribute to psychosocial distress [17]. The reported levels of mental illness in EVD survivors varies, with a systematic review by Lötsch et al. (2017) reporting prevalence of depression in EVD survivors ranging from 17 to 50% [12]. EVD survivors have also faced negative social, demographic, and economic outcomes, with prior research finding that survivors have experienced stigma, rejection by their community post-hospital discharge, permanent displacement, economic losses, and dissolution of their marriages [3, 11, 14, 15, 18, 19].

Although previous research has documented the impacts of EVD upon the physical and mental health of survivors, these studies have important limitations. Shortly after the epidemic, much of the research on the impacts of Ebola infection was qualitative, used small convenience samples, and typically did not include comparison groups of individuals who were not infected with EVD [8, 14, 15, 20–32]. Following much of the initial qualitative research, quantitative studies emerged that followed EVD survivors for one to five years after the outbreak to measure the impact of EVD on physical and mental health [4, 6, 7, 33, 34]. Among these studies, several did not include a control group and only measured health outcomes for EVD survivors over short periods since infection [7, 33, 34]. Other studies included comparison groups of individuals who lived with them at the time of infection or shortly afterwards. Although the inclusion of a control group is essential to account for health changes in the non-EVD population over time, a group that co-resided with Ebola-infected individuals likely overrepresents individuals affected by EVD, since they share many characteristics of the survivors. For example, since EVD primarily spread through close contacts, they have a relatively higher likelihood of having similar experiences of, e.g., grief, loss, reduction in wealth status, and, thus, potential of exhibiting physical and mental health symptoms. As a result, this may underestimate the effect of EVD on some outcomes compared to the general population. Therefore, knowledge of the long-term physical and mental health impacts on EVD survivors compared to the general population is still limited.

By surveying EVD survivors five to seven years after infection and using a population-based comparison group, we address the limitations of previous research here. To do so, we use population-based quantitative data from Kenema District, Sierra Leone, which includes

both EVD survivors and a comparison group of uninfected individuals from communities affected by EVD. These data were collected between October and November of 2021 in Kenema District. Using these data, our goal was to identify the long-term impact of EVD on physical and mental health outcomes of survivors.

## Methods

### Setting

The 2013–2016 West Africa epidemic began in the Mano River triangle at the intersection of Guinea, Liberia, and Sierra Leone, inhabited by the cross-border Kissi tribe [2]. The first case of EVD, the spillover event, was likely a small boy who had been playing in a tree inhabited by bats in Meliandou, Guinea in December 2013 [2]. Due to the fluidity of family borders and the ease of crossing borders by road, foot, and boat, the virus quickly spread to Sierra Leone and Liberia [2]. The first case of EVD in Sierra Leone was in May 2014 in Kailahun District [2]. In Sierra Leone, EVD cases were hidden for months after its arrival, in part due to mistrust and fear, emerging only after the virus was already propagating in Kailahun and Kenema Districts [35].

The Eastern Province, including Kenema District, served as an early epicenter of the outbreak in Sierra Leone. Many health care workers contracted EVD [36], followed by their family members, and the community at large, with a lab-confirmed case count of 497 and more suspected in Kenema District [37]. Cases in Kenema spanned from mid-2014 to 2015 before lulling to a stop, largely due to increased isolation measures and use of Ebola Treatment Units (ETUs) by affected communities [2, 35].

### Sampling and recruitment

In this study, EVD survivors were identified from the national registry of Ebola infections maintained by our partner, the Sierra Leone Association of Ebola Survivors (SLAES). This registry was created during the epidemic and lists all laboratory-confirmed Ebola cases in Eastern Province of Sierra Leone, verified by SLAES using ETU discharge certificates. The registry contained characteristics of EVD survivors, including age, sex, area of residence when infected, whether they died from infection, and full contact information (including location and phone number, if available). At the time of data collection, there were 164 EVD survivors in the SLAES-Kenema database (including 143 EVD survivors over 15 years of age and 21 EVD survivors under 15 years of age), making it one of the most affected districts in Sierra Leone.

EVD survivors were recruited by SLAES-Kenema executive board members. The entire population of EVD survivors over 15 years of age in Kenema District was contacted, and the resulting sample was an estimated 76.9% of eligible EVD survivors (110 out of 143).

Control group participants were selected as a representative sample of those exposed to EVD in Kenema district. For this group, we first obtained a list of all enumeration areas (EAs) within Kenema district from the 2020 Sierra Leone census. From this list, we included only EAs that had at least one EVD infection, according to the EVD infection database described above. Among the 107 EAs that had an infection, we selected eight EAs, with the selection probability proportionate to the number of EVD cases from the infection database. Within these selected EAs, we selected 40 households for a quantitative survey interview. To do so, we used the number of households in the EA from the 2020 census and obtained maps of the selected EAs. We then selected one household randomly as a starting point, and then selected every $n$th household after this starting point to obtain a final number of 40 households per EA. Within the household, we interviewed the male or female head (one person per household).

Participants were excluded from the study if they were under 15 years of age or if they did not live in Kenema District. EVD survivors were also excluded if they were not in the SLAES database. Otherwise, all EVD survivors and community members selected via the sampling frame were eligible to participate in the survey.

## Data collection

Participants were interviewed by twelve field researchers from Njala University, supervised by two members of the Kenema Government Hospital Outreach Team. Interviewers and supervisors attended a one-week training prior to data collection where they were taught the protocol and practiced interviewing participants. All interviews were conducted verbally in Krio or Mende due to low literacy rates in the target population. The survey was run on smart phones using the Open Data Collect app, and data was uploaded to the ONA server.

The team conducted regular data checks on a daily basis throughout data collection, examining items like potential interviewer effects, the number of interviews per day, and reports of sociodemographic characteristics. Any issues identified in these checks were reported to data collection supervisors to address in the field. There were no missing variables for the data used in this analysis.

**Measures.**   The survey instrument consists of sections concerning demographics, economics, fertility and contraception, marital history, sexual behavior, health, health care access, social capital, community, stigma, migration, and COVID-19 (S1 Appendix). Participants were asked about the current presence of physical and mental health signs and symptoms from a checklist consisting of signs and symptoms prevalent in EVD survivors. Our selection of symptoms measured in this study is based on similar previous studies [5, 6, 8, 9]. Participants who indicated that they were experiencing symptoms were subsequently asked about the duration (in years). They were also asked how much the symptom interferes with their lives, measured with a 3-point Likert scale: "does not interfere at all," "interferes a little," and "interferes a lot."

## Data analysis

De-identified data was analyzed in Stata, using a variety of summary statistics, comparison of means tests, and bivariate and multivariate regressions based on the nature of the questions and the dependent variables. All $p$-values are two-sided.

**Participant demographics.**   Demographic variables were analyzed for frequency of responses, and then for significant differences between the EVD survivor and community member groups through chi square analyses (gender, education, location marital status, ethnicity, religion, wealth status), or Mann-Whitney tests (age, number of children), depending upon the nature of the dependent variable (categorical or continuous, respectively).

Wealth was measured via proxy indicators due to the difficulties of collecting comparable data about the long-term economic statuses of households. Instead, participants were asked about their household assets, for example, presence of electricity or a radio in their home. These variables were utilized to generate Wealth Indices through a Principal Components Analysis, with a varimax rotation and extraction of eigenvalues greater than 1, retaining five components [38]. Cases were then ranked into equivalent tertiles using a method adapted from programs such as the Demographic Health Survey and World Food Program [38, 39]. This measure of wealth is relative to the rest of the dataset, categorizing participants with "higher," "middle," or "lower" wealth statuses.

**Health symptom analysis.**   Total symptom counts and separate physical and mental health symptom counts were analyzed with negative binomial regression models. This was

suitable, as the dependent variable consisted of count data, the independent variable and covariates were categorical or continuous, and observations were independent. Symptoms that had arisen before the EVD outbreak began, eight years prior to the survey, were excluded. McFadden's pseudo-$R^2$ was assessed to determine goodness of fit.

Health signs/symptoms questions were analyzed via binomial logistic regression models due to the binary nature of the independent variable of Ebola survivorship (*0 or 1*) and the dependent variable of the presence of each symptom (*0 or 1*). Data consisted of independent observations, with no multicollinearity of independent variables, no extreme outliers, a linear relationship between independent variables and the logit of the response variable, and a sufficiently large sample size. Models were assessed for goodness of fit using the Hosmer-Lewe-show test and retained only if p > 0.05.

Data regarding duration of symptoms and severity were obtained for each "yes" response for symptom presence. Duration of symptoms was analyzed with descriptive outputs and an independent t-test. Total ranking of symptom severity was analyzed with an ordinal regression due to the ordered Likert scale of responses (no/little/a lot of interference). The proportional odds assumption was tested, with insignificant results of the likelihood-ratio test of proportionality of odds across response categories (p > 0.05) and Brant test of the parallel regression assumption (p > 0.05). The goodness of fit of the ordinal regression model was measured with McFadden's pseudo-$R^2$. Frequency outputs were also run on total symptom interference ratings.

All multivariate regression models were adjusted for age, gender, and proxy indicators of socioeconomic status (SES) including wealth status, education level, marital status, and number of children still living due to prior association of the variables with the exposure of EVD and the outcomes of physical and mental statuses [7, 9, 40–45]. Covariates were assessed to confirm lack of multicollinearity using the variance inflation factor, and interactions between covariates were tested using interaction terms.

## Ethical compliance

Ethical clearance was obtained by the Institutional Review Boards (IRB) of Johns Hopkins University and Tulane University, the Sierra Leone Ethics Review Board, and the School of Global Health of University of Copenhagen. Data was collected and stored in accordance with the Principles of Helsinki and General Data Protection Regulation (GDPR) [46, 47].

Verbal informed consent was obtained from each study participant, in their own language (Krio or Mende), and was witnessed and documented by data collectors in Open Data Kit. Verbal consent was approved by the IRB as an alternative to written consent due to low literacy rates amongst the target population. Participants from 15 to 18 years of age provided assent, and one parent was required to provide consent for study participation. In accordance with local customs, consent was also obtained from the SLAES-Kenema executive board and the chiefs of entered communities.

Data was pseudo-anonymized. Coordinating identifiers were removed before analyzing the data, in accordance with GDPR. All data was stored on secure servers. Anonymized survey results were disseminated in forums of interested participants and study personnel, as well as stakeholders at the local, national, and international levels.

## Results

### Participant demographics

Demographic characteristics and differences between EVD survivor (*N* = 110) and non-EVD survivor community member (*N* = 319) survey participants are presented in Table 1.

**Table 1. Demographics of EVD survivor and community member survey participants, Kenema District, 2021.**

| Participant Demographics | EVD Survivors (*n/N*) | Community Members (*n/N*) | *p*-Value |
|---|---|---|---|
| **Gender** | | | 0.328 |
| Female | 66.4% (73/110) | 61.1% (195/319) | |
| Male | 33.6% (37/110) | 38.9% (124/319) | |
| **Age (median + IQR)** | 35.0 (24.0–42.0) | 30.0 (22.0–42.0) | 0.210 |
| 15–19 | 11.8% (13/110) | 11.9% (38/319) | |
| 20–29 | 25.5% (28/110) | 37.9% (121/319) | |
| 30–39 | 27.3% (30/110) | 20.4% (65/319) | |
| 40–49 | 20.9% (23/110) | 11.0% (35/319) | |
| 50–59 | 10.0% (11/110) | 8.8% (28/319) | |
| 60–69 | 0.9% (1/110) | 5.3% (17/319) | |
| 70–79 | 0.9% (1/110) | 3.4% (11/319) | |
| > 80 | 2.7% (3/110) | 1.3% (4/319) | |
| **Marital Status** | | | <0.001[a] |
| Married/living together | 44.5% (49/110) | 49.8% (159/319) | |
| Separated/divorced | 5.5% (6/110) | 6.0% (19/319) | |
| Widowed | 23.6% (26/110) | 4.7% (15/319) | |
| Never married | 26.4% (29/110) | 39.5% (126/319) | |
| **Number of Children (median + IQR)** | 3.0 (1.0–6.0) | 2.0 (0.0–4.0) | <0.001[a] |
| 0–3 | 51.8% (57/110) | 69.0% (220/319) | |
| 4–7 | 38.2% (42/110) | 24.4% (78/319) | |
| 8–11 | 7.3% (8/110) | 4.7% (15/319) | |
| > 11 | 2.7% (3/110) | 1.9% (6/319) | |
| **Location** | | | <0.001[a] |
| Rural | 66.4% (73/110) | 20.4% (65/319) | |
| Town | 5.5% (6/110) | 38.9% (124/319 | |
| City | 28.2% (31/110) | 40.8% (130/319) | |
| **Education Level** | | | 0.174 |
| None | 43.6% (48/110) | 32.0% (102/319) | |
| Primary | 8.2% (9/110) | 9.1% (29/319) | |
| Junior Secondary | 9.1% (10/110) | 9.7% (31/319) | |
| Senior Secondary | 23.6% (26/110) | 34.5% (110/319) | |
| Higher | 15.5% (17/110) | 14.6% (47/319) | |
| **Current Wealth Status** | | | 0.001[a] |
| Lowest | 41.8% (46/110) | 32.0% (102/319) | |
| Middle | 39.0% (43/110) | 30.1% (96/319) | |
| Highest | 19.0% (21/110) | 37.9% (121/319) | |
| **Ethnicity** | | | 0.466 |
| Mende | 76.4% (84/110) | 68.3% (218/319) | |
| Mandingo | 5.5% (6/110) | 6.6% (21/319) | |
| Krio | 0.9% (1/110) | 0.0% (0/319) | |
| Fullah | 3.6% (4/110) | 4.1% (13/319) | |
| Kissi | 2.7% (3/110) | 3.8% (12/319) | |
| Kono | 1.8% (2/110) | 2.5% (8/319) | |
| Sherbro | 2.7% (3/110) | 1.6% (5/319) | |
| Temne | 4.5% (5/110) | 5.6% (18/319) | |
| Other Sierra Leone | 1.8% (2/110) | 2.5% (8/319) | |
| Limba | 0.0% (0/110) | 4.1% (13/319) | |

*(Continued)*

**Table 1.** (Continued)

| Participant Demographics | EVD Survivors (*n/N*) | Community Members (*n/N*) | *p*-Value |
|---|---|---|---|
| Loko | 0.0% (0/110) | 0.3% (1/319) | |
| Other Foreign | 0.0% (0/110) | 0.6% (2/319) | |
| **Religious Affiliation** | | | 0.338 |
| Muslim | 73.6% (81/110) | 69.3% (221/319) | |
| Christian | 26.3% (29/110) | 30.7% (98/319) | |

[a]Statistically significant (p < 0.05).

Table 1 shows that the demographic compositions of the EVD survivor sample and the community member-based comparison group were relatively similar but demonstrated significant differences in marital status, number of children, location, and wealth status. Survivors had more children on average ($p < 0.001$), had less wealth ($p = 0.001$), and lived in more rural areas ($p < 0.001$). In addition, they were more likely to be widowed, at 23.6% vs. 4.7% ($p < 0.001$).

In both respondent groups, females were slightly more represented than men, at nearly a 1:2 ratio. Both respondent groups were Mende and Muslim by majority. Represented ages skewed younger and older in the comparison group, with a greater number of respondents in the 20–29 and 60–79 age ranges, and less in the 30–59 age ranges. While not significantly different, education levels also varied, with EVD survivors more likely to have no education (43.6%) than community members (32.0%).

## Signs and symptoms reported by Ebola survivors

A negative binomial regression was utilized to analyze the association of survivor status and the total number of reported symptoms present, and adjusted for confounders (age, gender, education, number of children, and wealth status). EVD survivors are 3.10 (95% CI, 2.63–3.65) times more likely to experience physical and mental health symptoms than members of the comparison group, a statistically significant result at $p < 0.001$. Unadjusted, the IRR was 3.18 (95% CI, 2.73–3.71). In total, EVD survivors experience a median of 4.0 (2.0–6.0) reported symptoms, whereas the comparison group had a median of 1.0 (1.0–2.0). 100.0% of EVD survivors and 85.6% of community members reported the presence of at least one health symptom.

When viewed separately, groups of physical and mental health symptoms were both more likely amongst EVD survivors than community members, with respective adjusted IRRs of 2.65 (95% CI, 2.28–3.09), $p < 0.001$, and 11.95 (95% CI, 6.58–21.71), $p < 0.001$. Adjusted IRRs were 2.67 (95% CI, 2.31–3.08) and 12.97 (95% CI, 7.38–22.80). 99% of EVD survivors and 79.0% of community members reported at least one physical health symptom, whereas 40% of EVD survivors and 5.0% of community members reported at least one mental health symptom.

## Self-reported physical and mental health symptoms

The comparative odds of EVD survivors ($N = 110$) experiencing EVD-related health symptoms versus non-EVD survivor community members ($N = 319$) are shown in Table 2.

Nearly all the physical and mental health signs and symptoms were statistically significant, exhibiting a much higher likelihood of occurring in EVD survivors. Some particularly survivor-specific physical health signs and symptoms were vision problems [AOR = 23.34 (95% CI,

**Table 2. Prevalence and odds ratios of self-reported signs and symptoms in EVD survivors and community members, Kenema District, 2021.**

| Signs/Symptoms | EVD Survivors (n/N) | Community Members (n/N) | Unadjusted OR (95% CI) | Adjusted[a] OR (95% CI) | p-Value |
|---|---|---|---|---|---|
| **Physical Health Symptoms** | | | | | |
| Weight loss | 11.8% (13/110) | 6.0% (19/319) | 2.12 (1.01–4.44) | 2.13 (0.98–4.64) | 0.057 |
| Fatigue/weakness | 31.8% (35/110) | 15.7% (50/319) | 2.51 (1.52–4.15) | 2.45 (1.45–4.12) | <0.001[b] |
| Headaches | 67.3% (74/110) | 43.3% (138/319) | 2.70 (1.70–4.25) | 2.89 (1.80–4.66) | <0.001[b] |
| Confusion | 24.5% (27/110) | 7.2% (23/319) | 4.19 (2.28–7.68) | 4.38 (2.31–8.30) | <0.001[b] |
| Appetite loss | 17.3% (19/110) | 8.2% (26/319) | 2.35 (1.24–4.45) | 2.21 (1.13–4.35) | 0.021[b] |
| Vision problems | 44.5% (49/110) | 3.4% (11/319) | 22.49 (11.07–45.71) | 23.34 (11.06–49.28) | <0.001[b] |
| Hearing loss | 12.7% (14/110) | 0.9% (3/319) | 15.36 (4.32–54.57) | 14.68 (3.98–54.18) | <0.001[b] |
| Heart problems | 26.4% (29/110) | 2.8% (9/319) | 12.33 (5.61–27.09) | 10.15 (4.46–23.11) | <0.001[b] |
| Difficulty breathing | 8.2% (9/110) | 1.6% (5/319) | 5.60 (1.83–17.08) | 7.93 (2.32–27.17) | <0.001[b] |
| Chest pain | 23.6% (26/110) | 3.8% (12/319) | 7.92 (3.83–16.35) | 9.09 (4.14–19.98) | <0.001[b] |
| Stomach pain | 29.1% (32/110) | 15.7% (50/319) | 2.21 (1.33–3.68) | 2.55 (1.47–4.41) | <0.001[b] |
| Diarrhea | 0.0% (0/110) | 1.3% (4/319) | - | - | - |
| Decreased libido | 3.6% (4/110) | 0.9% (3/319) | 3.97 (0.87–18.05) | 5.63 (1.10–28.72) | 0.038[b] |
| Joint pain | 75.5% (83/110) | 28.8% (92/319) | 7.58 (4.61–12.47) | 7.85 (4.65–13.25) | <0.001[b] |
| Numbness of extremities | 13.6% (15/110) | 1.3% (4/319) | 12.43 (4.03–38.36) | 9.61 (2.88–32.04) | <0.001[b] |
| **Mental Health Symptoms** | | | | | |
| Feeling suddenly scared for no reason | 13.6% (15/110) | 0.3% (1/319) | 50.21 (6.55–385.06) | 60.15 (7.51–481.78) | <0.001[b] |
| Loss of sexual interest or pleasure | 14.5% (16/110) | 1.9% (6/319) | 8.88 (3.38–23.33) | 8.49 (3.10–23.24) | <0.001[b] |
| Difficulty falling asleep or staying asleep | 20.0% (22/110) | 1.9% (6/319) | 13.04 (5.13–33.16) | 12.28 (4.67–32.26) | <0.001[b] |
| Feeling fearful | 0.0% (0/110) | 0.9% (3/319) | - | - | - |
| Spells of terror or panic | 25.5% (28/110) | 0.3% (1/319) | 108.58 (14.56–809.90) | 146.64 (18.43–1166.55) | <0.001[b] |
| Trembling | 3.6% (4/110) | 0.6% (2/319) | 5.98 (1.08–33.12) | 6.53 (1.06–40.28) | 0.043[b] |

[a] Adjusted for age, gender, education, marital status, number of children, and wealth status.

[b] Statistically significant ($p < 0.05$).

11.06–49.28), $p < 0.001$], hearing loss [AOR = 14.68 (95% CI, 3.98–54.18), $p < 0.001$], heart problems [AOR = 10.15 (95% CI, 4.46–23.11), $p < 0.001$], and numbness of extremities [AOR = 9.61 (95% CI, 2.88–32.04), $p < 0.001$]. Mental health signs and symptoms were also significantly higher in EVD survivors versus community members, particularly spells of terror and panic [AOR = 146.64 (95% CI, 18.43–1166.55), $p < 0.001$] and feeling suddenly scared for no reason [AOR = 60.15 (95% CI, 7.51–481.78), $p < 0.001$].

Survivors reported higher levels of each sign and symptom compared to community members, aside from two with relatively low levels (diarrhea and feeling fearful). The five most prevalent physical sequalae reported by EVD survivors were joint pain (75.5%), headaches (67.3%), vision problems (44.5%), fatigue/weakness (31.8%), and stomach pain (29.1%). The most prevalent psychological symptoms were spells of terror and panic (25.5%) and difficulty falling asleep or staying asleep (20.0%).

## Duration of symptoms

For those who reported symptoms present, they were also asked how many years it had been a problem for them. Survivors reported experiencing symptoms for a median of 6.0 (3.0–7.0) years, compared to a median of 2.0 (1.0–3.0) years for community members. A Mann-Whitney test determined that this was statistically significant difference, $p < 0.001$.

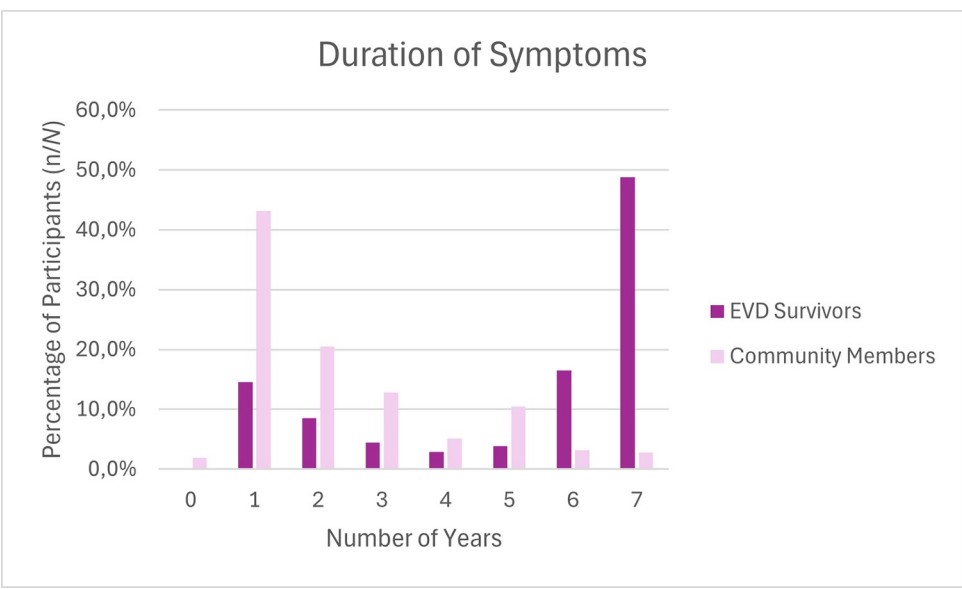

**Fig 1. Frequency distribution of duration of symptoms in EVD survivors and community members.**

The frequency distribution of responses for EVD survivors and community members also demonstrated skews in the opposite directions (left and right, respectively). EVD survivors are more likely to have experienced symptoms for six years (16.5%) or seven years (48.8%), around the time of the epidemic, whereas community group members were more likely to have experienced symptoms for one year (43.2%) or two years (20.5%) (see Fig 1).

## Symptom interference in life

Participants were also asked to rank how much the problem interferes with their lives. An ordinal regression was run to determine the effect of surviving EVD upon ratings of symptom interference in the lives of participants. EVD survivors were found to be 2.99 (95% CI, 2.27–3.93) times more likely to rank their symptoms as causing higher interference than community members after adjusting for age, gender, education, marital status, number of children, and wealth status, a statistically significant effect at $p < 0.001$. The unadjusted OR was 3.10 (95% CI, 2.40–4.00).

This is also evident in Fig 2, which illustrates that EVD survivors had a higher frequency of symptoms that they rated as causing interference in their lives. Survivors ranked 44.4% of their symptoms as causing a lot of interference, and only listed 7.6% of symptoms as causing no interference at all. Community members, on the other hand, had a relatively normal distribution, with the majority of symptoms (61.3%) causing a little interference.

## Discussion

In this study, we use quantitative population-based data from Kenema District, Sierra Leone to measure the long-term impacts of EVD on physical and mental health in survivors compared to a population-based sample of community members who lived in EVD-affected areas but were not diagnosed with EVD.

The findings strongly indicate that Ebola Virus Disease (EVD) survivors in Kenema District, Sierra Leone still have unmet health needs six years after the epidemic. They are experiencing disproportionately lower health than community members due to significantly

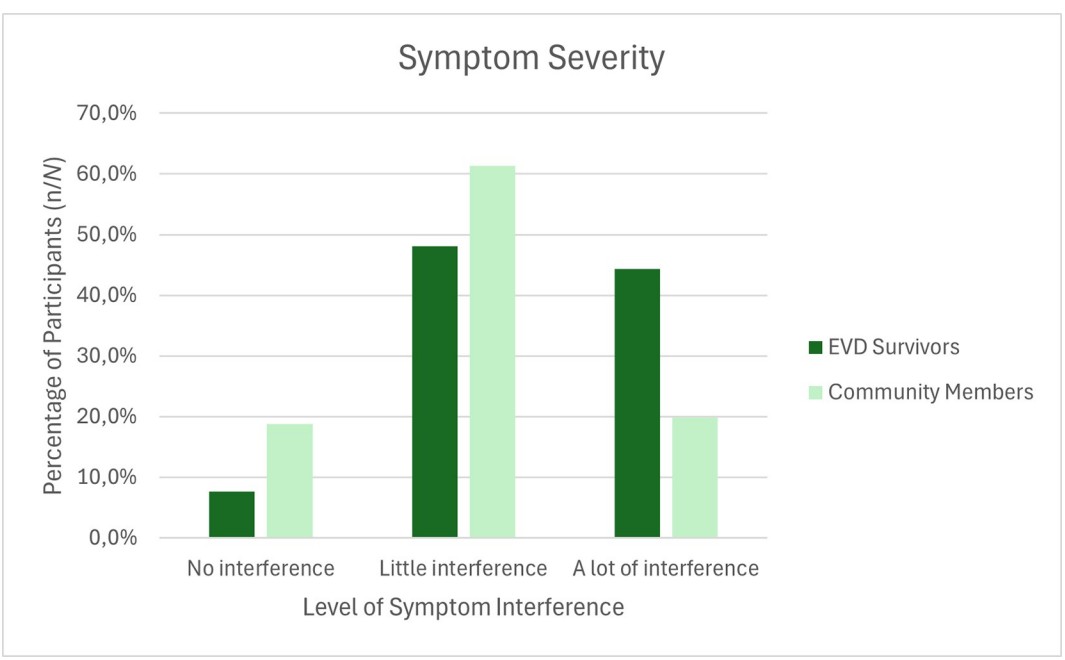

**Fig 2. Frequency distribution of symptom interference in life in EVD survivors and community members.**

higher numbers of post-EVD physical and mental health symptoms. Most of these symptoms have been present since the time of the infection or shortly thereafter, six and seven years prior to survey administration. Survivors were also significantly more likely to rank existing symptoms as causing high interference in their lives. The population-based findings that EVD survivors are still experiencing more physical and mental health symptoms than their community members are extremely compelling, especially given the length of time between the EVD epidemic and data collection.

Physical health symptoms reported by EVD survivors aligned with those reported by other studies, although many other studies were restricted to the first few years following the epidemic. Multiple large-scale cohort studies surveying self-reported physical sequalae of EVD survivors, including the Liberian Ebola Survivor Cohort and PREVAIL III in Liberia and Post-EboGui in Guinea, reported long-term impacts for a range of two to six years after infection [5–7, 9]. Comparability between studies is limited, as survey tools inquired about different symptoms, and only PREVAIL-III included a comparison group of close contacts who lived with EVD survivors at the time of diagnosis. However, they generally found some of the same common symptoms of joint pain, headaches, fatigue, and ocular issues.

These studies reported a range of symptom prevalences in their populations. At a median of six years post-discharge, the Liberian Ebola Survivor Cohort team reported joint pain (39.3%), headaches (32.5%), fatigue (30.7%), and vision problems (18.1%) as the most prominent symptoms in EVD survivors [8]. The PREVAIL-III study reported uveitis (33.3%), headaches (32.6%), joint pain (27.6%), cataracts (13.8%), and muscle pain (12.8%) as the most prevalent symptoms in EVD survivors two years after infection, and all were significantly more likely to occur in EVD survivors than close contacts [6]. Four years post-discharge, Post-EboGui found general symptoms (25.3%), abdominal pain (17.1%), musculoskeletal (16.8%), and ocular symptoms (6.1%) to be the most common symptoms experienced by EVD survivors [7]. Although this study identified common symptoms, we found higher levels of joint pain (75.5%), headaches (67.3%), vision problems (44.5%), fatigue/weakness (31.8%), and

stomach pain (29.1%) compared to other studies, even though our data was collected more than six years after the epidemic.

Ocular issues are specifically well-documented as a unique sequela in EVD survivors from the West Africa epidemic. Ocular issues were represented in the survey results as a significant concern, as 44.5% of survivors reported vision problems [AOR = 23.34 (95% CI, 11.06–49.28), $p$-value <0.001]. Prior literature has described scarring, uveitis leading to eye pain, redness, vision loss, and cataracts in EVD survivors, which often require specialty services at secondary, or even tertiary, facilities [48–51].

The few longitudinal studies that have been done have found mixed results of changes of symptoms over time. At baseline (an average of one year after acute EVD), the Liberian Ebola Survivor study found that 75.5% of EVD survivors reported one new cardinal physical health symptom since surviving EVD, which 85.8% rated as highly interfering with their lives [9]. However, five years after acute infection, all symptoms except numbness of feet and hands had significantly decreased. Only 52% of participants reported experiencing at least one symptom, and 29% of reported symptoms were classified highly interfering with their lives [9]. PostEbo-Gui and PREVAIL-III also noted decreases in sequelae in the years following infection [6, 7], aside from ocular complications, which increased over time [6]. As this study does not have a baseline it is impossible to compare, but it is notable that levels are still high, and significantly higher than that of community members, more than six years post-infection.

Findings of mental health signs and symptoms varied. Most anxiety- and PTSD-related symptoms listed in the survey were reported by survivors at significantly higher levels than community members. The most prevalent mental health symptom was spells of terror or panic, which was present in 25.5% of surveyed EVD survivors and 129.02 times (95% CI, 16.24–1024.67), $p < 0.001$, more likely to occur in EVD survivors than community members. EVD survivors also reported feeling suddenly scared for no reason, loss of sexual interest or pleasure, difficulty falling asleep or staying asleep, and trembling at significantly higher levels than community members. Marginal numbers of EVD survivors and community members reported feeling fearful, and the difference was not significant.

While we cannot directly compare these measures to other studies due to differences in scales and diagnostic criteria, these findings align with prior evidence describing long-term mental health impacts of EVD such as anxiety, depression, and PTSD [12]. A study conducted in 2017 with survivors of the 1995 EVD outbreak in Kikwit, Democratic Republic of Congo found significantly higher levels of depression and anxiety in EVD survivors compared to close contacts, 22 years after the epidemic [52]. Depression is the most documented mental illness in EVD survivors, with Lötsch et al. (2017)'s systematic review of long-term neuropsychological sequelae reporting levels of depression ranging from 17 to 50%, with a pooled total of 20% [12]. The authors noted that nearly all the included articles used different methods, some had relatively small samples, and few included comparison groups. Bah et al. (2020) surveyed 197 EVD survivors using the HADs scale in Bombali District, Sierra Leone in 2017 and found levels of 24.9% for anxiety, 47.2% for depression, and 21.8% for PTSD [53]. While this study does not have directly comparable results, it was clear that anxiety and PTSD-related mental health symptoms were far more likely to occur in EVD survivors than community members [AOR = 11.95 (95% CI, 6.58–21.71), $p < 0.001$], aligning with prior research about the neuropsychological impacts of EVD.

Some of the variations between reported levels of symptoms may be due to the cultural differences in wording and experiences of mental health, which could be explored through subsequent research. A formative report based on interviews with EVD survivors in Bombali and Kenema Districts described that survivors often discussed anxieties or constant worries, describing these feelings as "worried," "stressed," "embittered," "sad," "angry," "disturbed,"

and "frustrated" [54]. Utilizing surveys previously validated and adapted to similar populations and cultural contexts may provide more accurate insight into the mental health concerns of EVD survivors.

Furthermore, effective treatment of chronic conditions such as joint pain and ocular issues is limited in Sierra Leone, as in many parts of West Africa, due to limited resources and conflicting priorities [55–57]. The mental health system is also weak, with a lack of human resources and no national budget line for mental health [19]. While EVD survivors are still beneficiaries of the Free Health Care Initiative, prior research has suggested that this is inadequate to address their health concerns [58], and that access to and quality of care is dependent upon the strength of their local health facilities.

There are numerous strengths of the study, primarily being that it was population-based and had a comparison group sampled from the communities of EVD survivors. The response rate was high at 76.9% of EVD survivors over 15 years of age in Kenema District, and those who were not surveyed were not notably different in age and gender compared to those who were. The study is likely to be generalizable to the experiences of EVD survivors in Kenema District and similar areas impacted by the West Africa outbreak, with some applicability to other EVD outbreaks in similar settings and similar epidemics in West Africa.

## Limitations and bias

Selection bias, while limited, may have occurred to some extent. While recruitment by the SLAES Executive Board allowed the team to survey the majority of EVD survivors in Kenema District, those who were not included due to prior death or relocation may have offered different perspectives. In addition, the EVD survivors surveyed were in the SLAES database, were laboratory-verified cases, and had certificates. There may be others who were infected with EVD but did not get tested and go to an ETU, and are therefore not included in the survey.

We also note that symptoms were self-reported and not verified by a physician or measured with objective biological markers. While there are strengths to capturing the perceptions of survivors concerning their health status, these perceptions are subject to recall and information bias. EVD survivors may have responded cautiously to potentially sensitive and stigmatizing topics such as mental health and sexual dysfunction due to desirability bias, whereas they may have been inclined to over-report health concerns due to the dynamics of humanitarian aid and the history of individuals receiving funding based on disclosing their health status [59]. In addition, while the survey primarily focused on their present experience of symptoms, the question regarding the duration of symptoms is subject to recall bias.

As the study design was cross-sectional, causality cannot be established. This is an observational, descriptive study of the perceptions of EVD survivors about experiences with post-EVD sequelae compared to their community members. However, the study had strengths in establishing associations between survivorship and health outcomes due to its robust, population-based sample and comparison group.

## Conclusions

Findings from this study indicate that EVD survivors require a comprehensive, long-term package of care built into existing health care systems with a strong consideration of sustainability. Many EVD survivor programs ceased to exist after international attention, and thus funding declined, leaving survivors suddenly unable to access prior sources of care. Given the increasing threat of emerging infectious diseases, including recent outbreaks of COVID-19, EVD in DRC and Uganda, and Marburg in Equatorial Guinea, it is highly relevant to consider

the long-term impacts of diseases upon the health status and social determinants of health (including socioeconomic status) of survivors.

Given the finding that EVD survivors are significantly more likely to experience more physical and mental health signs and symptoms of increased severity more than six years after the epidemic, it is clear that epidemic responses need to integrate a longer-term, health systems strengthening and health equity-based approach to providing care for survivors. In addition, further research should be undertaken to determine health care-seeking behaviors of EVD survivors and availability of health care services to identify potential gaps and areas that could be strengthened through international and national programming and guidelines.

## Supporting information

**S1 Checklist. Inclusivity in global research.**
(DOCX)

**S1 Appendix. Abbreviated survey tool.**
(DOCX)

**S1 Data. Health survey data.**
(XLSX)

## Author Contributions

**Conceptualization:** Jenna N. Bates, Lina Moses, Donald S. Grant, Philip Anglewicz.

**Data curation:** Jenna N. Bates.

**Formal analysis:** Jenna N. Bates.

**Funding acquisition:** Lina Moses, Philip Anglewicz.

**Investigation:** Jenna N. Bates, Abdulai Kamara, Mohamed Sheku Bereteh, Denise Barrera, Lina Moses, Philip Anglewicz.

**Methodology:** Jenna N. Bates, Lina Moses, Philip Anglewicz.

**Project administration:** Jenna N. Bates, Abdulai Kamara, Mohamed Sheku Bereteh, Denise Barrera, Lina Moses, Allieu Sheriff, Fudia Sesay, Mohamed S. Yillah, Philip Anglewicz.

**Resources:** Lina Moses, Allieu Sheriff, Fudia Sesay, Mohamed S. Yillah, Donald S. Grant, Joseph Lamin, Philip Anglewicz.

**Software:** Joseph Lamin.

**Supervision:** Lina Moses, Donald S. Grant, Philip Anglewicz.

**Validation:** Lina Moses, Philip Anglewicz.

**Writing – original draft:** Jenna N. Bates.

**Writing – review & editing:** Jenna N. Bates, Lina Moses, Philip Anglewicz.

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
