## [Decision Letter · Decision Letter 0]

3 Apr 2024

PGPH-D-24-00262

Long-term physical and mental health outcomes of Ebola Virus Disease survivors in Kenema District, Sierra Leone: a cross-sectional survey

Dear Dr. Jenna Nicole Bates,,

Thank you for submitting your manuscript to PLOS Global Public Health. After careful consideration, we feel that it has merit but does not fully meet PLOS Global Public Health’s publication criteria as it currently stands. Therefore, we invite you to submit a revised version of the manuscript that addresses the points raised during the review process. Reviewer 1 has additional comments in the attached document

EDITOR Comments :

1. Please provide a clear study justification given that a lot of studies have been conducted on this topic. what new knowledge is the study providing

2. The reviewers were concerned about how the symptoms were measured and the how the data was statistically analysed. I suggest you pay close attention to that as this can be serious ground for rejection. 

3. Another issue that need to be addressed  is why the research team decided  to use community members as comparison group. why was close family members of EVD survivors were considered 

We look forward to receiving your revised manuscript.

Kind regards,

Peter Bai James, PhD

Academic Editor

Journal Requirements:

1. In the ethics statement in the Methods, you have specified that verbal consent was obtained. Please provide additional details regarding how this consent was documented and witnessed, and state whether this was approved by the IRB.

2. Please include a complete copy of PLOS’ questionnaire on inclusivity in global research in your revised manuscript. Our policy for research in this area aims to improve transparency in the reporting of research performed outside of researchers’ own country or community. The policy applies to researchers who have travelled to a different country to conduct research, research with Indigenous populations or their lands, and research on cultural artefacts. The questionnaire can also be requested at the journal’s discretion for any other submissions, even if these conditions are not met.  Please find more information on the policy and a link to download a blank copy of the questionnaire here: https://journals.plos.org/globalpublichealth/s/best-practices-in-research-reporting. Please upload a completed version of your questionnaire as Supporting Information when you resubmit your manuscript.

3. Please update your online Competing Interests statement. If you have no competing interests to declare, please state: “The authors have declared that no competing interests exist.”

4. In the online submission form, you indicated that your data will be submitted to a repository upon acceptance. We strongly recommend all authors deposit their data before acceptance, as the process can be lengthy and hold up publication timelines. Please note that, though access restrictions are acceptable now, your entire minimal dataset will need to be made freely accessible if your manuscript is accepted for publication. This policy applies to all data except where public deposition would breach compliance with the protocol approved by your research ethics board. If you are unable to adhere to our open data policy, please kindly revise your statement to explain your reasoning and we will seek the editor's input on an exemption.

5. Please ensure that you refer to Table 2 in your text as, if accepted, production will need this reference to link the reader to the table.

Additional Editor Comments (if provided):

Reviewers' comments:

Reviewer's Responses to Questions

**Comments to the Author**

1. Does this manuscript meet PLOS Global Public Health’s publication criteria? Is the manuscript technically sound, and do the data support the conclusions? The manuscript must describe methodologically and ethically rigorous research with conclusions that are appropriately drawn based on the data presented.

Reviewer #1: Yes

Reviewer #2: Partly

2. Has the statistical analysis been performed appropriately and rigorously?

Reviewer #1: Yes

Reviewer #2: No

3. Have the authors made all data underlying the findings in their manuscript fully available (please refer to the Data Availability Statement at the start of the manuscript PDF file)?

Reviewer #1: No

Reviewer #2: No

4. Is the manuscript presented in an intelligible fashion and written in standard English?

Reviewer #1: Yes

Reviewer #2: Yes

5. Review Comments to the Author

Reviewer #1: There is a need to provide more insights into the finding. The presence of over dispersion in the poisson regression model is a concern and is due to the limited data that was used in this study..however this can be handled by a quasi poisson regression. also the demographic statistics are two basic. so much were mentioned in the methodology section about the Test statistics used in this study but less is Test statistics results were presented.

Reviewer #2: Review PGPH-D-24-00262

Abstract

2013-2016 epidemic

Nb cases: more than 2 8000 (cf introduction, ref 1 ; [1] WHO Ebola Response Team, After Ebola in West Africa — Unpredictable Risks, Preventable Epidemics, N. Engl. J. Med. 375

(2016) 587–596.)

Limited prior research : replace by a few studies

Introduction

- Ebola is no longer clasified as an hemmorrhagic fever by WHO: prefer EVD;

- "2014-2016 EVD epidemic": change to "2013-2016 EVD epidemic";

- 14124 cases = 8706 confirmed + 287 probable + 5131 suspected

- CFR: nb of deaths in SL = 3956 but 3589 among 8706 confirmed  CFR among confirmed cases = 41%  to be precised

- "The vast majority of research on the impacts of Ebola infection is qualitative and used convenience samples". True for some studies looking at psychological consequences but not true given large survivors cohorts, for example:

. Guinea/Postebogui cohort (n=802): Etard in ref 4 ; Diallo in ref 36.

. Sierra Leone/Kerry Town ETC (n=151): K. Wing, S. Oza, C. Houlihan, J.R. Glynn, S. Irvine, C.E. Warrell, A.J.H. Simpson, S. Boufkhed, A. Sesay, L. Vandi, S.C.

Sebba, P. Shetty, R. Cummings, F. Checchi, C.R. McGowan, Surviving Ebola: A historical cohort study of Ebola mortality and survival in Sierra Leone 2014-2015, PLoS One. 13 (2018) 1–17.

https://doi.org/10.1371/journal.pone.0209655; H. Bower, E. Smout, M.S. Bangura, O. Kamara, C. Turay, S. Johnson, S. Oza, F. Checchi, J.R. Glynn, Deaths, late deaths, and role of infecting dose in Ebola virus disease in Sierra

Leone: retrospective cohort study, BMJ. (2016). https://doi.org/10.1136/bmj.i2403.

. Liberia/PREVAIL III (n=966) in ref 5 ; Longitudinal Liberian Ebola Survivor Study (n=299) in L. Overholt, D.A. Wohl, W.A. Fischer, D. Westreich, S. Tozay, E. Reeves, K. Pewu, D. Adjasso, D. Hoover, C. Merenbloom, H. Johnson, G. Williams, T. Conneh, J. Diggs, A. Buller,

D. McMillian, D. Hawks, K. Dube, J. Brown, Stigma and Ebola survivorship in liberia: Results from a longitudinal cohort study, PLoS One. 13 (2018) 1–13. https://doi.org/10.1371/journal.pone.0206595.

- "In addition, much of the quantitative research on the social and demographic impacts of Ebola was conducted with small samples of less than 120 participants...": ref ?

- "research has only examined the impacts of Ebola infection occurring shortly after recovery.": wrong, cf: prospective Postebogui cohort S.M.K. Diallo, A. Toure, M.S. Sow, C. Kpamou, A.K. Keita, B. Taverne, M. Peeters, P. Msellati, T.A. Barry, J.F. Etard,

R. Ecochard, E. Delaporte. Understanding Long-term Evolution and Predictors of Sequelae of Ebola Virus Disease Survivors in Guinea: A 48-month prospective, longitudinal cohort study (PosteboguI), Clin. Infect. Dis. 73 (2021)

2166–2174. https://doi.org/10.1093/cid/ciab168.

Methods

- "The 2014-2016 West Africa epidemic...": change to 2013-

- "the virus quickly spread to Sierra Leone and Liberia [29].": EVD spread in Kailahun district in eastern Sierra Leone in May 2014 - replace ref 29 (article in a general press newspaper, The NYT) by WHO Ebola Response Team, After Ebola in

West Africa — Unpredictable Risks, Preventable Epidemics, N. Engl. J. Med. 375 (2016) 587–596.

- Sampling.

. SLAES database: only confirmed cases in the database resulting in 176 confirmed cases in Kenema ? 143 vs 176: difference due to exclusion of age < 15 years ?

. Controls: sample size calculation is missing; I understand the sampling frame is a 2-stage PPS with first stage selection of 8 EA proportional to the number of survivors in each EA and at the second stage the selection

of a fixed number of HH (40) per EA and one individual per HH selected ; questions : what is the total number of EAs in Kenema from 8 EAs were selected ? for the 2d stage, what was the field procedure to select the 40 HH, apart saying

it was systematic ? and how was selected the unique participant from the HH, which procedure did you follow ? Clarify the overall sampling procedures.

. Measures: overall, more details are needed to assess the quality of the collected information on the symptoms; is the questionnaire available on-line ? how did you define and diagnose heart problems ? how did you assess duration of symptoms, given the fact it is very difficult in this setting to collect accurate answers on

dates ? how did you define severity/symptoms interferring with life ? how did you assess hearing and vision deficiencies ?

- Data anlaysis: describe data management, data ckeck procedures, missing data handling.

- Wealth/data reduction: it seems you retained one component and categorized it in 3 levels after the PCA but the PCA should have led to several components (eigenvalues > 1) ? Did you select the first component ?

- Analytic methods:

. "A Poisson regression model was selected due to the right skew of the dependent variable, ...": the dependent variable is a count (nb of symptoms) which is the main reason to use a Poisson model.

. "assessed with the Pearson dispersion statistic": did you assess it ? a negative binomial distribution could have been more adequate ?

. "Data consisted of independent observations, with no multicollinearity of independent variables, no extreme outliers, a linear relationship between independent variables and the logit of the response variable, and a

sufficiently large sample size": any use of goodness-of-fit statistics ?

Results

- Demo (ref to Table 1):

. The period (range of dates ) of discharge from ETC of the included survivors should be given.

. Precise in the text the number of participants in both groups.

. "Differences between demographics of the EVD survivor and community member participant groups were calculated with independent t-tests and chi square tests, depending on

the nature of the dependent variable. T-tests were run to compare the means of age and number of children due to their continuous nature, whereas chi square 213 tests were run for categorical dependent variables.": move to methods.

. Table 1: give the context in the title(study pop, location, year); some p-values are not in the right column there is a borderline difference in age distribution <60/60+ between survivors and controls (60+ : 4.5% vs 10.0%; Fisher exact p=0.05).

. "Differences in ages and genders may reflect limitations of the sampling methodology. Due to the survey data collection in the harvest period at the beginning of the dry season (October-November 2021) [46], middle-aged men

may have been working at their farms, while younger and older individuals (primarily female) may have been at home. Gender differentials may also match the demographics of the EVD survivor sample due to similar gender norms, as females are thought to be more likely to be

infected with EBOV due to their traditional role as caretakers [35].": there is no difference in gender in table 1 but this paragraph should be moved to the Discussion section.

. "Variances in location....Therefore, it cannot be used as a reliable measure of location." : move to discussion.

- Sequelae (ref to Table 2):

. "They were asked to select...number of children, and wealth status).": move to discussion.

. I suggest to start the paragraph with the grouped symptoms aand the Poisson regression (lines 257-260).

- Self-reported...

. "In addition, binomial logistic regressions...to generate ORs and AORs.": move to methods.

. "spells of terror and panic [AOR = 134.09 (95% CI, 17.41-1032.74), p <0.001] and feeling suddenly scared for no reason [AOR = 64.27 (95% CI, 8.08-511.03), p <0.001]." : very large CI due to only one occurence in the

control group, an indication of the goodness of fit is necessary.

. Figure 1 gives the same information and less accurately than Table 2.

Discussion

- Very long: more than 6 pages, the discussion on mental health consequences while very important could be shorten.

- "Unlike previous research, this study has a comparison group": PREVAIL III included a comparison group.

- "although many other studies were restricted to the first few years following the epidemic" : the clinical follow-up in Guinea/Postebogui was 4 years (Ref 36).

- "Liberian Ebola Survivor Cohort": not in the References list (see above).

- Ocular issues : add Ref E.H. Hebert, M.O. Bah, J.F. Etard, M.S. Sow, S. Resnikoff, C. Fardeau, A. Toure, A.N. Ouendeno, I.C. Sagno, L. March, S. Izard, P.L. Lama, M. Barry, E. Delaporte. Ocular Complications in Survivors of

the Ebola Outbreak in Guinea, Am. J. Ophthalmol. 175 (2017) 114–121. https://doi.org/10.1016/j.ajo.2016.12.005.

- Strenghs of the study: a population-based comparison is truely a major strengh and should be recognized ; it's a clear difference with other studies which used contacts as a comparison group ; it's a good reason to detail the sampling methodology.

- "and those who were not surveyed were not notably different in age and gender compared to those who were." : not documented.

- Limitations.

. to further discuss selection bias, give the period of discharge, closed to dates of admission in ETC/ETU (see above) ? did recruitment uptake in the survivor cohort change over time during the outbreak ?

. "... the EVD survivors surveyed were laboratory-verified cases and had certificates": info to be added in the Methods section.

. "... responses were all self-reported and not verified by a physician": to be discussed in relation to recall bias and information bias.

. "... with a high statistical power due to its robust, population based sample and comparison group": power is not assessed, so the point is purely declarative.

. a comment on the absence of any objective biological markers should be added.

Conclusion: yes. I agree on a survivors care package available through the current health system after the end of the international aid. It falls within the global stenghthening of the health system of the low-income countries.

6. PLOS authors have the option to publish the peer review history of their article (what does this mean?). If published, this will include your full peer review and any attached files.

**Do you want your identity to be public for this peer review?** For information about this choice, including consent withdrawal, please see our Privacy Policy.

Reviewer #1: No

Reviewer #2: No

---

## [Decision Letter · Decision Letter 1]

30 Aug 2024

PGPH-D-24-00262R1

Long-term physical and mental health outcomes of Ebola Virus Disease survivors in Kenema District, Sierra Leone: a cross-sectional survey

Dear Dr.
Jenna Nicole Bates,

Thank you for submitting your manuscript to PLOS Global Public Health. After careful consideration, we feel that it has merit but does not fully meet PLOS Global Public Health’s publication criteria as it currently stands. Therefore, we invite you to submit a revised version of the manuscript that addresses the points raised during the review process.

EDITOR COMMENTS: Please insert comments here and delete this placeholder text when finished. Be sure to:

1. Please share the data that informed the findings of this study. PLOS journals mandate that authors must publicly share all data essential for replicating their study's results without any restrictions at the time of publication. If legal or ethical constraints prevent the public dissemination of a dataset, authors are required to detail the process for others to gain access to the data. 

PLOS highly advocates for the use of data repositories whenever feasible. These repositories enhance the discoverability and accessibility of data, guarantee its preservation over time, and contribute to heightened visibility for the associated research.

Please submit your revised manuscript by the **13th September 2024** If you will need more time than this to complete your revisions, please reply to this message or contact the journal office at globalpubhealth@plos.org. Please include the following items when submitting your revised manuscript:

We look forward to receiving your revised manuscript.

Kind regards,

Peter Bai James, PhD

Academic Editor

Journal Requirements:

1. In the ethics statement in the Methods, you have specified that verbal consent was obtained. Please provide additional details regarding how this consent was documented and witnessed, and state whether this was approved by the IRB

2. Please include a complete copy of PLOS’ questionnaire on inclusivity in global research in your revised manuscript. Our policy for research in this area aims to improve transparency in the reporting of research performed outside of researchers’ own country or community. The policy applies to researchers who have travelled to a different country to conduct research, research with Indigenous populations or their lands, and research on cultural artefacts. The questionnaire can also be requested at the journal’s discretion for any other submissions, even if these conditions are not met.  Please find more information on the policy and a link to download a blank copy of the questionnaire here: https://journals.plos.org/globalpublichealth/s/best-practices-in-research-reporting. Please upload a completed version of your questionnaire as Supporting Information when you resubmit your manuscript.

Additional Editor Comments (if provided):

Reviewers' comments:

Reviewer's Responses to Questions

**Comments to the Author**

1. If the authors have adequately addressed your comments raised in a previous round of review and you feel that this manuscript is now acceptable for publication, you may indicate that here to bypass the “Comments to the Author” section, enter your conflict of interest statement in the “Confidential to Editor” section, and submit your "Accept" recommendation.

Reviewer #3: All comments have been addressed

Reviewer #4: All comments have been addressed

2. Does this manuscript meet PLOS Global Public Health’s publication criteria? Is the manuscript technically sound, and do the data support the conclusions? The manuscript must describe methodologically and ethically rigorous research with conclusions that are appropriately drawn based on the data presented.

Reviewer #3: Yes

Reviewer #4: Yes

3. Has the statistical analysis been performed appropriately and rigorously?

Reviewer #3: Yes

Reviewer #4: Yes

4. Have the authors made all data underlying the findings in their manuscript fully available (please refer to the Data Availability Statement at the start of the manuscript PDF file)?

Reviewer #3: No

Reviewer #4: Yes

5. Is the manuscript presented in an intelligible fashion and written in standard English?

Reviewer #3: Yes

Reviewer #4: Yes

6. Review Comments to the Author

Reviewer #3: Overall Comment

This is an interesting paper on the long term health outcomes of Ebola survivors in Sierra Leone. The authors convey the case study in the language of EVD evolution, and multilevel factorial dynamics characterised by iteration and emergence. While the paper is an interesting contribution to the literature, the revised version has two main weaknesses. Firstly, in terms of methods, and nature of ‘data collection’ and secondly in terms of empirical/description detail.

Firstly, the methodology is unclear and in some sub-sections contradictory. The data collection reads as qualitative, I doubt the use of the term ‘interviews’ as the appropriate data collection approach used in this study, a study that is quantitatively analyzed and presented. Forms of interviews, discussions, and observations are qualitative data collection methods/approaches and therefore should be analysed as such, thematically, rounded-theory etc…

On the other hand, if the data collection of this study involved reading out surveys to participants and reporting their responses in the survey sheet, that is not an interview, it is a standard survey approach for participants who cannot read. So, I would urge that, if the data collection strategy involved distribution and completion of a survey, as it appears to be, then do not refer to that process as interviews are done.

However, if actual interviews were conducted, then you would need to provide a description of the type of interviews done, suitability for this study, theme/interview guides, variations of participants, heterogeneity, positionality/ and qualitative analysis plan. This would also nullify the entire result presentation in this study. The analysis, whether grounded or thematic should be explained.

In a mixed-methods approach, which may involve both interviews (qualitative) and Survey quantitative) the interpretation of both data and synergies in both sets of findings be presented and discussed.

Secondly, on the empirical/description basis of the physical symptoms: A clear distinction of symptoms/signs and sequelae of EVD should be made. Were these symptoms/signs noted/reported during EVD and therefore have progressed to chronicity (now Sequelae) or were these symptoms/signs intermittent occurrences? I see these terms used interchangeably throughout the manuscript but would be good to differentiate. Symptoms like fever and diarrhoea can be mostly considered acute and or intermittent precipitated by an ongoing process. It would be a big claim to consider the persistence of such symptoms for 7 years after the initial EVD infection.

The paper presents limited evidence of how the study’s findings are different from similarly studies done in 5 years period after EVD occurrence. Would an additional year (6 or 7) compared to that of 5-year studies, identify new symptoms or simply track progression/improvement of the same symptoms studied throughout the previous 5 years? And how does the local health system structure to address expected age-related conditions such as joint pain etc, access to symptomatic treatment and specialist services for chronic sequalae in the discussion section.

I hope these comments are useful and encourage the authors to further strengthen this interesting piece.

Specific Comments

1. Page 4, Line 79: The statement ‘…with joint and muscle pain, fever, headaches….’ should not report fever as sequelae.

2. Page 7, Line 160: clarify whether “interview” refers to qualitative interviews, as data collection. Please change and check throughout for consistency.

3. Page 7, Line 160-164: is this the Kish grid approach, and household selection strategy? How feasible is this in Sierra Leone and Kenema in identifying the specific household?

4. Page 8, Line 183: "Measures". A table of symptoms and sequelae, definitions and criteria for inclusion as EVD cause would improve this sub-section.

5. Page 9, Line 194: Did you use both statistical software for your analysis or just one (which one)?

6. Page 9, Line 205: The phrase “ownership of electricity” is not clear. Please rephrase.

7. Page 9, Line 213: How was missing data and multicollinearity handled.

8. Page 20, Line 399: Change ‘six -years’ to ‘more than Six-year’

9. Page 21, Line 432: The statement ‘This is an interesting finding’, could rephrased more specifically. Is ‘this’ referring to all sequelae previously listed or one in particular?

10. Reference 3 may be incomplete, please review

11. Please review references 37 and 57

12. Figure 'Duration of Symptoms' could be presented differently. Grouping all symptoms (one or more) and plotting that to number of participants could be misleading. I wound not expect fever to be present for 6+ years but Joint pain might be which could be non-EVD related. Different acute/intermittent symptoms and Chronic sequelae, to be plot with participants.

Reviewer #4: Overall, the authors have addressed the reviewers' concerns adequately.

The authors' data sharing statement suggests that data will be shared upon acceptance. This should be placed into a repository and shared.

7. PLOS authors have the option to publish the peer review history of their article (what does this mean?). If published, this will include your full peer review and any attached files.

**Do you want your identity to be public for this peer review?** For information about this choice, including consent withdrawal, please see our Privacy Policy.

Reviewer #3: **Yes: **Mamadu Baldeh

Reviewer #4: No

---

## [Decision Letter · Decision Letter 2]

8 Oct 2024

Long-term physical and mental health outcomes of Ebola Virus Disease survivors in Kenema District, Sierra Leone: a cross-sectional survey

PGPH-D-24-00262R2

Dear Dr
Jenna Nicole Bates

We are pleased to inform you that your manuscript 'Long-term physical and mental health outcomes of Ebola Virus Disease survivors in Kenema District, Sierra Leone: a cross-sectional survey' has been provisionally accepted for publication in PLOS Global Public Health.

Best regards,

Peter Bai James, PhD

Academic Editor

Reviewer Comments (if any, and for reference):

Reviewer's Responses to Questions

**Comments to the Author**

1. If the authors have adequately addressed your comments raised in a previous round of review and you feel that this manuscript is now acceptable for publication, you may indicate that here to bypass the “Comments to the Author” section, enter your conflict of interest statement in the “Confidential to Editor” section, and submit your "Accept" recommendation.

Reviewer #3: All comments have been addressed

2. Does this manuscript meet PLOS Global Public Health’s publication criteria? Is the manuscript technically sound, and do the data support the conclusions? The manuscript must describe methodologically and ethically rigorous research with conclusions that are appropriately drawn based on the data presented.

Reviewer #3: Yes

3. Has the statistical analysis been performed appropriately and rigorously?

Reviewer #3: Yes

4. Have the authors made all data underlying the findings in their manuscript fully available (please refer to the Data Availability Statement at the start of the manuscript PDF file)?

Reviewer #3: Yes

5. Is the manuscript presented in an intelligible fashion and written in standard English?

Reviewer #3: Yes

6. Review Comments to the Author

Reviewer #3: Thank you for addressing the comments satisfactorily.

Short comment to consider: While the term 'interview' as a data collection tool for a study type or not is rooted in philosophical and methodological discussions, an approach where the investigator and participant operate as independent entities, examining a phenomenon without mutual influence—wherein "inquiry occurs through a one-way direction", 'necessary' is not regarded as an 'interview.'

Qualitative "interviews" are grounded on interpretivism and constructivism through conversational techniques, contrasting with the positivist paradigm of quantitative instruments designed to analyze causal and associative relationships independently of human perspectives.

Ideally, the term 'interview' widely meets a qualitative approach. To prevent conflict of data collection tools, the term 'standardized' reduces the 'interviewers' effects' in quantitative tools to that of a qualitative study.

7. PLOS authors have the option to publish the peer review history of their article (what does this mean?). If published, this will include your full peer review and any attached files.

**Do you want your identity to be public for this peer review?** For information about this choice, including consent withdrawal, please see our Privacy Policy.

Reviewer #3: No
